# Different contributions of fat and lean indices to bone strength by sex

**Yen-Huai Lin**[1,2]*, **Michael Mu Huo Teng**[1,2]

**1** Department of Medical Imaging, Cheng Hsin General Hospital, Taipei, Taiwan, **2** Department of Medicine, School of Medicine, National Yang Ming Chiao Tung University, Taipei, Taiwan

* yhlin11@gmail.com

**Data Availability Statement:** The data set is owned by the IRB of Cheng Hsin General Hospital. The IRB of Cheng Hsin General Hospital only approved the data analysis in our study and did not approve data sharing. Therefore, we do not have permission to share the data set. Interested

## Abstract

Bone strength depends on both bone density and quality. However, the differences in how body composition indices affect bone strength between men and women remains unclear. This study investigated the associations of various fat and lean indices with bone strength by sex. In this cross-sectional study involving 1,419 participants, bone mineral density (BMD) and body composition were measured using dual-energy X-ray absorptiometry. Bone quality was assessed using the trabecular bone score (TBS). Fat indices included total fat mass, body fat percentage, and waist circumference, while lean indices included appendicular lean mass (ALM) and hand grip strength. All fat indices demonstrated a positive association with BMD and a negative association with the TBS in both men and women. Fat indices were more strongly associated with BMD in women than in men. Furthermore, lean indices contributed more to BMD in men than in women. In women, ALM contributed more to BMD than hand grip strength, whereas in men, hand grip strength had a greater impact on BMD than ALM. Hand grip strength was also positively associated with the TBS in men. Overall, fat indices had a greater influence on BMD in women, while lean indices were more positively associated with bone strength in men. Considering different fat indices, ALM was more strongly associated with BMD in women, whereas hand grip strength played a greater role in men. Thus, maintaining both muscle mass and strength is crucial for preserving bone mass.

## Introduction

The World Health Organization defines obesity as abnormal or excessive fat accumulation that may impair health and recommends body mass index (BMI) as a simple tool to evaluate obesity [1]. BMI in this context is used worldwide; however, it does not distinguish between fat and lean mass [2]. In recent years, technological improvements have enabled the use of dual-energy X-ray absorptiometry (DXA) to assess obesity by measuring the body fat percentage (BFP) [3]. Additionally, android adiposity is associated with more health problems than gynoid adiposity, which can be easily determined by waist circumference (WC) or waist-to-hip ratio calculation [4]. Different methods of evaluating obesity may result in discordant

**Funding:** This work was supported by a grant (CHGH113-N23) from the Cheng Hsin General Hospital, Taipei, Taiwan. There was no additional external funding received for this study.

**Competing interests:** The authors have declared that no competing interests exist.

classifications. Shah and Braverman showed that 48% of women and 22% of men not classified as having obesity based on BMI were found to have obesity based on BFP [5].

Previous studies have reported an association of a higher BMI or weight with better bone mineral density (BMD) because of the higher skeletal load [6–8]. The concept of the obesity paradox mainly depends on a simple obesity definition based on BMI; however, other diagnostic methods are available for determining obesity. The association between different fat indices and BMD has not yet been sufficiently explored. Kim et al. reported that BMI was positively associated with BMD, whereas BFP showed a negative association [9]. Yoo et al. showed that BFP was not associated with BMD in post-menopausal women [10]. Kim et al. reported that BFP and WC were positively associated with BMD [11]. Another study involving 6,776 metabolically healthy adults demonstrated that BMI and BFP were significantly associated with low bone mass [12]. Therefore, results regarding the association between different fat indices and BMD remain inconclusive. Furthermore, muscle mass and strength both play an important role in the lean indices [13,14], and muscle strength has been demonstrated to be associated with BMD [15,16]. However, these studies did not consider the effect of muscle strength on the association between fat indices and BMD [9–12].

The National Institutes of Health defined osteoporosis as a skeletal disease characterized by compromised bone strength, which is the integration of bone density and quality [17]. The International Society for Clinical Densitometry (ISCD) recommends that bone quality be measured by the trabecular bone score (TBS) [18]. Looker et al. reported a negative association of the TBS with various body size variables; however, their study did not incorporate fat and lean indices simultaneously and did not investigate the effect of muscle strength [19].

Body composition indices varied according to sex, and muscle strength was a key measure of lean indices. Bone strength was defined as the integration of bone density and quality. To the best of our knowledge, no previous study has comprehensively investigated the associations between various fat and lean indices with bone strength in men and women. Therefore, we aimed to elucidate the effects of the lean and fat indices on bone strength in men and women.

## Materials and methods

### Participants

This cross-sectional study was conducted between January 2018 and April 2024. Participants who underwent DXA examinations at our hospital were enrolled in this study. The inclusion criteria were men aged ≥50 years and women who were post-menopausal, both with a BMI between 15 and 37 kg/m$^2$. The exclusion criteria were as follows: inability to perform the hand grip strength (HGS) test or inability to cooperate with anthropometric measurements due to physical or cognitive disabilities. Ultimately, 1,419 participants were included. This study was approved by the institutional review board (IRB no. (660)107A-32) and the procedures used adhered to the tenets of the 1964 Declaration of Helsinki and its later amendments. All participants provided written informed consent.

### Anthropometric measurements

Participants wore light clothing during weight and height measurements, which were taken using an automatic height measurement and weighing scale instrument (HW-3030, Superview, Taoyuan, Taiwan). The WC was measured at the approximate midpoint between the top of the iliac crest and the lower margin of the last palpable rib.

## Muscle strength measurements

HGS was assessed using a digital hand dynamometer (EH101; Camry, Zhongshan, China). The grip strength of both hands was measured three times per hand, and the maximum value from each hand was considered for the analysis [14].

## Total fat mass, body fat percentage, and appendicular lean mass measurements

Body composition was analyzed using a DXA scanner (Horizon W; Hologic Inc., Bedford, MA, USA) to evaluate the BFP, total fat mass, and appendicular lean mass (ALM).

## Bone mineral density

BMD measurements were performed in accordance with the official position of the ISCD from 2019 [20]. The lumbar spine and both hips of the participants were evaluated using a DXA scanner (Horizon W; Hologic Inc.). The BMDs of the lumbar spine, femoral neck, and total femur were measured. The coefficients of variation of the BMDs were 1.28%, 1.96%, and 1.07% for the lumbar spine, femoral neck, and total hip, respectively.

## Bone quality and trabecular bone score

Bone quality was evaluated using the TBS as recommended by the ISCD. The TBS is a non-invasive method of evaluating bone quality based on DXA images [21]. We used iNsight software for the TBS measurements (version 3.0.2.0, Medimaps, Geneva, Switzerland). The coefficient of variation of the TBS was 2.00%.

## Other covariates

The covariates included age, dairy product intake (yes vs. no), regular exercise for at least 30 minutes a day on more than 3 days per week (yes vs. no), and medications affecting bone strength (yes vs. no). Medications affecting bone strength included bisphosphonates, parathyroid hormone, estrogens, and corticosteroids [15].

## Statistical analysis

All data analyses were performed using IBM SPSS Statistics for Windows (version 19.0; IBM Corp., Armonk, NY, USA). Independent $t$-tests or $\chi^2$ tests were used to compare the demographic characteristics between men and women. Multiple linear regression analysis was used to estimate the associations of fat and lean indices with BMD or the TBS. Multicollinearity diagnostics were performed in all models and assessed using variance inflation factors (VIF) with the same guide; values greater than 10 suggested a high degree of multicollinearity. An interaction model was built to investigate the statistical interaction between body composition indices and sex. A post-hoc power analysis based on an alpha level (0.05) was performed to strengthen the statistical robustness. For all analyses, a two-sided $p < 0.05$ was considered statistically significant.

## Results

The demographic characteristics of the participants were shown in Table 1. Women tended to have a lower mean age, lower mean BMI, lower WC, higher BFP, higher total fat mass, lower ALM, lower HGS, lower regional BMDs, and a lower TBS than men. Considering these differences, further analyses based on sex were performed.

**Table 1. Comparison of characteristics between men and women (n = 1,419).**

| Variables | Men (n = 319) | | Women (n = 1,100) | | p value for independent t/χ2 |
|---|---|---|---|---|---|
| | Mean | Standard deviation | Mean | Standard deviation | |
| Age (years) | 68.0 | 8.8 | 64.2 | 8.7 | <0.001 |
| Anthropometric measurements | | | | | |
| Weight (kg) | 68.9 | 10.8 | 57.9 | 9.5 | <0.001 |
| Height (cm) | 167.1 | 6.4 | 155.7 | 5.7 | <0.001 |
| Body mass index (kg/m$^2$) | 24.6 | 3.4 | 23.9 | 3.8 | 0.002 |
| Waist circumference (cm) | 88.7 | 10.3 | 80.4 | 9.4 | <0.001 |
| Body composition analysis | | | | | |
| Body fat percentage (%) | 34.3 | 5.0 | 43.2 | 5.0 | <0.001 |
| Total fat mass (kg) | 23.9 | 6.3 | 25.3 | 6.3 | <0.001 |
| Appendicular lean mass (kg) | 18.0 | 2.8 | 12.3 | 2.0 | <0.001 |
| Hand grip strength (kg) | 33.8 | 7.7 | 21.3 | 4.4 | <0.001 |
| Bone mineral density (g/cm$^2$) | | | | | |
| Lumbar spine | 0.946 | 0.18 | 0.819 | 0.15 | <0.001 |
| Femoral neck | 0.702 | 0.13 | 0.615 | 0.10 | <0.001 |
| Total hip | 0.877 | 0.15 | 0.761 | 0.11 | <0.001 |
| Trabecular bone score | 1.329 | 0.09 | 1.277 | 0.09 | <0.001 |
| Lifestyle factors, n (%) | | | | | |
| Exercise | 119 (37.3) | | 364 (33.1) | | 0.162 |
| Dairy product intake | 122 (38.2) | | 421 (38.3) | | 0.993 |
| Medications | 33 (10.3) | | 183 (16.6) | | 0.006 |

Multiple regression analyses were performed to investigate the contributions of lean and fat indices to BMDs at all sites and to the TBS. Total fat mass and ALM were analyzed in model 1, BFP and ALM in model 2, and WC and ALM in model 3. The value of $R^2$ represents the explanatory power of the model. In all models, total fat mass, BFP, WC, and ALM showed positive associations with BMDs at all sites, but they had negative associations with the TBS in women (Table 2). After HGS was adjusted in the model, the results were similar and the HGS had minimal association with total hip BMD (Table 3). VIFs in all models were less than 10, indicating a low degree of multicollinearity. The $R^2$ values in women for spine BMD and the TBS were low; however, they were high for both femoral neck and total hip BMDs. The $R^2$ did not change obviously, even after adjusting for HGS in the models.

In men, ALM was more positively associated with BMD than fat indices at all sites, whereas fat indices were negatively associated with the TBS (Table 4). The standardized coefficients (β) between fat indices and BMDs at all sites were greater in women than men, e.g., 0.219 vs. 0.117 in model 1 for spine BMD and 0.203 vs. 0.109 in model 1 for femoral neck BMD (Tables 2 and 4). Conversely, the standardized coefficients between ALM and BMD in all models were greater in men, e.g., 0.109 vs. 0.300 in model 1 for spine BMD and 0.185 vs. 0.330 in model 1 for femoral neck BMD (Tables 2 and 4). After adjusting for HGS in the models, HGS was observed to contribute more to BMD than ALM and most fat indices, e.g., 0.241 vs. 0.157 vs. 0.153 in model 1 for spine BMD (Table 5). HGS was more positively associated with BMDs and the TBS in men than in women. In contrast, HGS showed minimal association with total hip BMD in women (Table 3). Additionally, there was significant interaction between HGS and sex. The VIFs in all models were less than 10, indicating a low degree of multicollinearity. As with women, the $R^2$ values in men for spine BMD and the TBS were low, but they were high for both femoral neck and total hip BMDs. In contrast, when models were adjusted for

**Table 2. Multiple regression standardized coefficients (β) of various fat indices and appendicular lean mass in women.**

|  | Spine BMD | Femoral neck BMD | Total hip BMD | Trabecular bone score |
|---|---|---|---|---|
| Model 1 |  |  |  |  |
| Total fat mass | 0.219*** | 0.203*** | 0.274*** | -0.105** |
| Appendicular lean mass | 0.109** | 0.185*** | 0.162*** | -0.038 |
| $R^2$ | 0.142 | 0.213 | 0.217 | 0.274 |
| Model 2 |  |  |  |  |
| Body fat percentage | 0.120*** | 0.120*** | 0.174*** | -0.112*** |
| Appendicular lean mass | 0.221*** | 0.288*** | 0.299*** | -0.084** |
| $R^2$ | 0.124 | 0.200 | 0.196 | 0.279 |
| Model 3 |  |  |  |  |
| Waist circumference | 0.163*** | 0.152*** | 0.224*** | -0.172*** |
| Appendicular lean mass | 0.144*** | 0.213*** | 0.191*** | -0.006 |
| $R^2$ | 0.125 | 0.195 | 0.193 | 0.282 |

All regression models were adjusted for age, medications, exercise, and dairy product intake.

*$p<0.05$

**$p<0.01$

***$p<0.001$.

HGS, the $R^2$ values were higher in men than in women. A post-hoc power analysis showed that the power exceeded 80%.

## Discussion

This study investigated the difference in associations of fat and lean indices with bone strength according to sex. The results showed that fat indices, such as total fat mass, BFP, and WC, were

**Table 3. Multiple regression standardized coefficients (β) of various fat and lean indices including appendicular lean mass and hand grip strength in women.**

|  | Spine BMD | Femoral neck BMD | Total hip BMD | Trabecular bone score |
|---|---|---|---|---|
| Model 1 |  |  |  |  |
| Total fat mass | 0.210*** | 0.202*** | 0.275*** | -0.105** |
| Appendicular lean mass | 0.111** | 0.171*** | 0.142*** | -0.052 |
| Hand grip strength | 0.038 | 0.043 | 0.067* | 0.058 |
| $R^2$ | 0.145 | 0.208 | 0.216 | 0.287 |
| Model 2 |  |  |  |  |
| Body fat percentage | 0.113*** | 0.117*** | 0.174*** | -0.108*** |
| Appendicular lean mass | 0.219*** | 0.275*** | 0.281*** | -0.099*** |
| Hand grip strength | 0.033 | 0.039 | 0.063* | 0.055 |
| $R^2$ | 0.128 | 0.194 | 0.195 | 0.291 |
| Model 3 |  |  |  |  |
| Waist circumference | 0.159*** | 0.161*** | 0.238*** | -0.169*** |
| Appendicular lean mass | 0.144*** | 0.193*** | 0.164*** | -0.016 |
| Hand grip strength | 0.035 | 0.047 | 0.071* | 0.042 |
| $R^2$ | 0.133 | 0.195 | 0.198 | 0.297 |

All regression models were adjusted for age, medications, exercise, and dairy product intake.

*$p<0.05$

**$p<0.01$

***$p<0.001$.

**Table 4. Multiple regression standardized coefficients (β) of various fat indices and appendicular lean mass in men.**

|  | Spine BMD | Femoral neck BMD | Total hip BMD | Trabecular bone score |
|---|---|---|---|---|
| Model 1 |  |  |  |  |
| Total fat mass | 0.117 | 0.109 | 0.182** | -0.279*** |
| Appendicular lean mass | 0.300*** | 0.330*** | 0.318*** | 0.102 |
| $R^2$ | 0.135 | 0.210 | 0.231 | 0.117 |
| Model 2 |  |  |  |  |
| Body fat percentage | 0.056 | 0.064 | 0.119* | -0.224*** |
| Appendicular lean mass | 0.358*** | 0.384*** | 0.409*** | -0.033 |
| $R^2$ | 0.127 | 0.205 | 0.221 | 0.107 |
| Model 3 |  |  |  |  |
| Waist circumference | 0.151* | 0.080 | 0.183** | -0.216** |
| Appendicular lean mass | 0.281*** | 0.345*** | 0.317*** | 0.066 |
| $R^2$ | 0.142 | 0.206 | 0.232 | 0.098 |

All regression models were adjusted for age, medications, exercise, and dairy product intake.

*$p < 0.05$

**$p < 0.01$

***$p < 0.001$.

positively associated with BMD and negatively associated with the TBS in both men and women. Fat indices contributed more to BMD in women, whereas lean indices were more positively associated with BMD in men. The relative contribution of the lean indices to BMD was greater in men than in women. Additionally, lean indices had different effects on BMD between men and women. ALM was more strongly associated with BMD than HGS in

**Table 5. Multiple regression standardized coefficients (β) of various fat and lean indices including appendicular lean mass and hand grip strength in men.**

|  | Spine BMD | Femoral neck BMD | Total hip BMD | Trabecular bone score |
|---|---|---|---|---|
| Model 1 |  |  |  |  |
| Total fat mass | 0.153* | 0.143* | 0.224*** | -0.252*** |
| Appendicular lean mass | 0.157* | 0.180* | 0.136* | -0.046 |
| Hand grip strength | 0.241** | 0.246** | 0.315*** | 0.267*** |
| $R^2$ | 0.158 | 0.229 | 0.273 | 0.155 |
| Model 2 |  |  |  |  |
| Body fat percentage | 0.085 | 0.091 | 0.152** | -0.201*** |
| Appendicular lean mass | 0.238*** | 0.255*** | 0.251*** | -0.171* |
| Hand grip strength | 0.229** | 0.237** | 0.303*** | 0.275*** |
| $R^2$ | 0.148 | 0.222 | 0.259 | 0.147 |
| Model 3 |  |  |  |  |
| Waist circumference | 0.188** | 0.116 | 0.229*** | -0.188** |
| Appendicular lean mass | 0.134 | 0.196** | 0.131 | -0.092 |
| Hand grip strength | 0.246** | 0.241** | 0.315*** | 0.281*** |
| $R^2$ | 0.168 | 0.223 | 0.275 | 0.140 |

All regression models were adjusted for age, medications, exercise, and dairy product intake.

*$p < 0.05$

**$p < 0.01$

***$p < 0.001$.

women, whereas HGS was more strongly associated with BMD than ALM in men. Thus, the fat and lean indices contributed differently to bone health in men and in women.

The fat indices of total fat mass, BFP, and WC were revealed to be positively associated with BMDs at all sites in women. The explanatory power was higher for femoral neck and total hip BMDs than for spine BMD. Kim et al. demonstrated similar results, reinforcing the findings of our study [11]. In contrast, positive associations of total fat mass with BMDs at all sites, BFP with total hip BMD, and WC with spine and total hip BMDs was observed in men. Similar to the results for women, the explanatory power was higher for femoral neck and total hip BMDs than spine BMD in men. The standardized coefficients between fat indices and BMDs at all sites were greater in women than men, indicating that fat indices contributed more to BMD in women.

We further explored the association between the lean indices and BMD. Lean mass has been previously demonstrated to be a strong determinant of BMD in both men and women [11,22]. Furthermore, ALM was found to have a stronger association with BMDs at all sites in men than in women. Recently, HGS has come to be the key measure of sarcopenia [13,14] and has been demonstrated to be associated with BMD [15,16]. However, the effect of muscle strength on BMD has not been fully investigated [11,22]. After including ALM and HGS as lean indices in a multivariable model, we found different associations of ALM and HGS with BMD. ALM was more strongly associated with BMD in women, whereas HGS was more strongly associated with BMD in men. Additionally, the explanatory power did not change markedly with and without the adjustment for HGS in women, whereas it was higher with the adjustment for HGS in men. Therefore, lean indices, particularly HGS, were more strongly associated with BMD in men than in women. These results may be explained by the innate higher percentage of lean muscle mass in men than in women [23]; moreover, lower HGS has been more strongly associated with osteoporosis in men than in women [24].

Bone strength reflects the integration of bone density and bone quality [17]. The TBS is a DXA-derived variable recommended for measuring bone quality [18]. We found that all fat indices were negatively associated with the TBS, and the standardized coefficients were higher for WC in women and for total fat mass in men. However, the lean indices showed inconsistent results. ALM was negatively associated with the TBS in both men and women, whereas HGS was positively associated with the TBS in men only. Looker et al. also reported a negative association of the TBS with various body size variables [19]. Correlations with the TBS were significantly higher for WC but did not differ from the correlation for total body or trunk fat mass [19]. Interestingly, HGS was found to be positively associated with the TBS in men only. This may be because lean indices contributed more to BMD in men than women and the contraction of the muscle at its attachment to the bone provides a direct stimulus to maintain bone health [25,26].

The relationship between body composition and bone strength is complex. Fat mass and lean mass contribute to BMD because of mechanical loading [11]. Despite the positive association between fat mass and BMD, accumulating evidence supports the association between obesity and compromised bone quality. This is attributable to complex mechanisms, such as oxidative stress, inflammation, alteration of bone-regulating hormones, and dysfunction of the endocannabinoid system [27]. Therefore, the positive effects of obesity on BMD cannot offset its detrimental effects of obesity on bone quality [27]. In our study, we found that ALM was positively associated with BMDs at all sites in both men and women, even with various fat indices incorporated into the models. The relative contribution of ALM to BMD was greater in men than in women. These findings suggested that ALM may predict BMD regardless of variations in fat indices. In other words, maintaining ALM is important for BMDs at all sites in both men and women, regardless of fat mass or proportion. Additionally, HGS was more

associated with BMD and the TBS in men than women, even after incorporating ALM within multivariable models. Therefore, maintaining muscle mass and strength are equally important in men.

The different contributions of fat and muscle to bone health in men and women are primarily driven by biological mechanisms related to mechanical loading and the distinct ways in which fat and muscle interact with bone tissue. Muscle has a mechanostat function, in which muscles apply stress to bones through contractions during movement, triggering bone remodeling. This mechanical loading is detected by osteocytes, which then signal for new bone formation (osteogenesis) to reinforce the skeletal structure [25,26]. Men typically have greater muscle mass, leading to greater mechanical loading on bones. This results in a stronger stimulus for bone formation and an increased BMD. In contrast, women tend to have a lower muscle mass, which results in less mechanical loading and a weaker stimulus for bone remodeling. This can contribute to relatively lower BMD, particularly after menopause, when muscle mass may further decline. The role of fat includes mechanical loading from body weight. Like muscle mass, increased fat mass exerts mechanical loading on bones, stimulating bone formation [11]. Although this effect is less direct and efficient than muscle-induced mechanical loading, body weight from fat still contributes to bone health. Men generally have less body fat than women, leading to less mechanical loading from fat. In contrast, higher body fat in women contributes more significantly to mechanical loading and may positively influence BMD. However, excessive fat can have negative effects, as it may cause systemic inflammation [27]. The complex interplay between muscle and fat affects bone health differently in men and women, with hormonal differences modulating the effect of fat and muscle on bone health [25,26].

Our study had several strengths. First, we considered both lean indices, such as muscle mass and strength, and widely used fat indices, which are practical for clinical application. Second, we investigated the association between body composition variables and bone strength, including bone density and quality. Nevertheless, our study had some limitations that must be considered when interpreting the results. First, this was a cross-sectional study; therefore, the causal inferences could not be made. Second, the participants belonged exclusively to the Asian population, which may have affected the generalizability of our results. Third, our study only included men aged $\geq 50$ years and post-menopausal women; therefore, our results may not be applicable to younger men or premenopausal women. Further studies across various populations were needed to confirm our results.

## Conclusion

The fat and lean indices had different contributions to bone health between men and women. Considering different fat indices, ALM remained more strongly associated with BMD in women, whereas HGS made a stronger contribution to BMD in men. Thus, maintaining both muscle mass and strength is crucial for preserving bone mass. These results can help elucidate the effects of the lean and fat mass on bone strength in middle-aged and older adults of both sexes. Further longitudinal studies are required to investigate how changes in muscle mass or strength contribute to bone mass, independent of body size.

## Author Contributions

**Conceptualization:** Yen-Huai Lin.

**Data curation:** Yen-Huai Lin.

**Formal analysis:** Yen-Huai Lin.

**Funding acquisition:** Yen-Huai Lin.

**Investigation:** Yen-Huai Lin.

**Methodology:** Yen-Huai Lin.

**Project administration:** Yen-Huai Lin, Michael Mu Huo Teng.

**Resources:** Yen-Huai Lin, Michael Mu Huo Teng.

**Software:** Yen-Huai Lin.

**Supervision:** Yen-Huai Lin.

**Validation:** Yen-Huai Lin, Michael Mu Huo Teng.

**Visualization:** Yen-Huai Lin.

**Writing – original draft:** Yen-Huai Lin.

**Writing – review & editing:** Yen-Huai Lin, Michael Mu Huo Teng.

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
