## [Decision Letter · Decision Letter 0]

26 Jun 2024

PONE-D-24-20088The sex-differential association of fat and lean indices with bone strengthPLOS ONE

Dear Dr. Lin,

Thank you for submitting your manuscript to PLOS ONE. After careful consideration, we feel that it has merit but does not fully meet PLOS ONE’s publication criteria as it currently stands. Therefore, we invite you to submit a revised version of the manuscript that addresses the points raised during the review process.

We look forward to receiving your revised manuscript.

Kind regards,

Kiyoshi Sanada, PhD

Academic Editor

PLOS ONE

Journal Requirements:

This work was supported by a grant (CHGH113-N23) from the Cheng Hsin General Hospital, Taipei, Taiwan.

Reviewers' comments:

Reviewer's Responses to Questions

**Comments to the Author**

1. Is the manuscript technically sound, and do the data support the conclusions?

Reviewer #1: No

Reviewer #2: Partly

2. Has the statistical analysis been performed appropriately and rigorously? 

Reviewer #1: No

Reviewer #2: No

3. Have the authors made all data underlying the findings in their manuscript fully available?

Reviewer #1: No

Reviewer #2: Yes

4. Is the manuscript presented in an intelligible fashion and written in standard English?

Reviewer #1: No

Reviewer #2: No

5. Review Comments to the Author

**Reviewer #1:** Dear Author:

The study provides valuable insights into the sex-differential effects of fat and lean indices on bone strength. There are still some issues need to be clarified.

1.The study only includes men aged ≥50 years and post-menopausal women, which limits the applicability of the findings to younger men and premenopausal women. The materials are not pertinent! What is the study design rationale?

2.The study adjusts for several confounders but may still be subjected to residual confounding. Is it possible to utilize advanced statistical methods such as propensity score matching or structural equation modeling to further control for potential confounding variables.

3.The study reports positive associations between fat indices and bone density, and negative associations with bone quality, but does not thoroughly explore these relationships. In discussion, please provide a more detailed analysis of the results, or discuss the biological mechanisms that might explain the differential effects of fat and lean indices on bone strength.

**Reviewer #2: **The authors summarize the effects of fat and lean body mass on bone strength in men and women in this paper. It is considered to be interesting, containing new findings and elements that influence the resolution of the reader's questions. However, there are major problems with the overall structure and development of the paper. It is difficult to see the overall narrative and it is not clear what you want to argue. The explanation for the claim is insufficient. There are similar previous studies on the keywords presented. Therefore, the structure must be changed to claim novelty. Also, some results are not shown (e.g. Table 5). These indicate several major problems, which require significant modification. Please check the uploaded file.

6. PLOS authors have the option to publish the peer review history of their article (what does this mean?). If published, this will include your full peer review and any attached files.

Reviewer #1: **Yes: **Robert Wen-Wei, Hsu

Reviewer #2: No

---

## [Author Response · Author response to Decision Letter 0]

24 Jul 2024

Reply: Thank you for your comment. We have revised the manuscript to meet the style requirements.

This work was supported by a grant (CHGH113-N23) from the Cheng Hsin General Hospital, Taipei, Taiwan.

Reply: Thank you for your comment. We have added the statement “There was no additional external funding received for this study” to the funding statement on page 18, line 279. This funding statement has also been included in our resubmission cover letter.

Page 18, line 279.

Funding: This work was supported by a grant (CHGH113-N23) from the Cheng Hsin General Hospital, Taipei, Taiwan. There was no additional external funding received for this study.

Reply: Thank you for your comment. The dataset is owned by the IRB of Cheng Hsin General Hospital. The IRB approved the data analysis for our study but did not approve data sharing. Therefore, we do not have permission to share the dataset. Interested researchers can request access to the data by contacting the IRB of Cheng Hsin General Hospital at chghirb@chgh.org.tw. Access to the data will be granted in the same manner as it was for the authors, who do not have any special access privileges. We have included the data availability statement on page 18, line 286.

Page 18, line 286

Data Availability statement: 

The data set is owned by the institutional review board of Cheng Hsin General Hospital. The institutional review board of Cheng Hsin General Hospital only approved the data analysis in our study and did not approve data sharing. Therefore, we do not have permission to share the data set. Interested researchers can submit data access requests to the institutional review board of Cheng Hsin General Hospital through the following email address: chghirb@chgh.org.tw Others would be able to access the data in the same manner as the authors. 

Reply: Thank you for your comment. We have included the ethics statement in the methods section on page 6, line 105.

Page 6, line 105

This study was approved by the institutional review board (IRB no. (660)107A-32) and the procedures used adhered to the tenets of the 1964 Declaration of Helsinki and its later amendments. All participants provided written informed consent. 

Comments to the Author

1. Is the manuscript technically sound, and do the data support the conclusions?

Reviewer #1: No

Reviewer #2: Partly

Reply: Thank you for your comment. In the introduction, we stated that BMI is used worldwide but has limitations, such as not differentiating between fat and lean mass. We noted that there are other methods for evaluating obesity. We also mentioned that previous studies investigating the association between fat indices and BMD have been inconclusive, with few considering the effect of lean indices. Additionally, bone strength integrates bone density and quality, but the association between different fat indices and bone quality has not been examined in prior studies. Therefore, the aim of this study was to investigate the association between various fat and lean indices with bone strength, including bone density and quality, in both men and women.

In the results, we found a positive association between fat indices and bone density and a negative association with bone quality. Lean indices contributed more to BMD in men than in women, particularly through muscle strength. To the best of our knowledge, no previous study has reported the different contributions of fat and lean indices to bone strength.

In the discussion, we further explored the positive relationship between fat indices and bone density and the negative relationship with bone quality. We also discussed the results of our multiple regression analysis and provided the biological mechanisms that might explain the differential effects of fat and lean indices on bone strength.

In response to your insightful comments, we have revised the manuscript accordingly. Additionally, our manuscript has undergone another round of English editing to improve clarity and understanding. We hope that the revised manuscript will have an enhanced scientific impact.

2. Has the statistical analysis been performed appropriately and rigorously?

Reviewer #1: No

Reviewer #2: No

Reply: Thank you for your comment. Our research goals primarily involve adjusting for confounders to estimate direct relationships between variables, which aligns well with the capabilities of multiple regression. This method allows us to address our research questions straightforwardly while adhering to the assumptions and data structure of our study. In the discussion section, we not only delve deeper into the results of the multiple regression analysis but also provide our interpretations. Our responses to the reviewers’ comments are provided below, and the revised manuscript has been updated accordingly. We hope that the revised manuscript will have an enhanced scientific impact.

3. Have the authors made all data underlying the findings in their manuscript fully available?

Reviewer #1: No

Reviewer #2: Yes

Reply: Thank you for your comment. The dataset is owned by the IRB of Cheng Hsin General Hospital. The IRB approved the data analysis for our study but did not approve data sharing. Therefore, we do not have permission to share the dataset. Interested researchers can request access to the data by contacting the IRB of Cheng Hsin General Hospital at chghirb@chgh.org.tw. Access to the data will be granted in the same manner as it was for the authors, who do not have any special access privileges. We have included this information in the data availability statement on page 18, line 286.

Page 18, line 286

Data Availability statement: 

The data set is owned by the institutional review board of Cheng Hsin General Hospital. The institutional review board of Cheng Hsin General Hospital only approved the data analysis in our study and did not approve data sharing. Therefore, we do not have permission to share the data set. Interested researchers can submit data access requests to the institutional review board of Cheng Hsin General Hospital through the following email address: chghirb@chgh.org.tw Others would be able to access the data in the same manner as the authors. 

4. Is the manuscript presented in an intelligible fashion and written in standard English?

Reviewer #1: No

Reviewer #2: No

Reply: Thank you for your comment. Our manuscript has undergone an additional round of English editing to enhance the language and improve clarity and understanding.

5. Review Comments to the Author

Reviewer #1: Dear Author:

The study provides valuable insights into the sex-differential effects of fat and lean indices on bone strength. There are still some issues need to be clarified.

1.The study only includes men aged ≥50 years and post-menopausal women, which limits the applicability of the findings to younger men and premenopausal women. The materials are not pertinent! What is the study design rationale?

Reply: Thank you for your comment. Our study included only men aged 50 years and older and post-menopausal women for several reasons. First, the study aimed to evaluate osteoporosis and body composition simultaneously among participants who underwent DXA examinations at our hospital. According to the 2019 ISCD Adult Official Positions, osteoporosis may be diagnosed in men aged 50 years and older and post-menopausal women (Ref). Therefore, our inclusion criteria focused on these groups. Second, our study was hospital-based rather than a population survey, making it rare for younger men and premenopausal women to undergo DXA examinations. Consequently, our results may not be generalizable to younger men or premenopausal women, which we have noted as a limitation on page 17, line 264.

(Ref) The International Society for Clinical Densitometry (2019) ISCD Official Positions—Adult. https://iscd.org/learn/official-positions/adult-positions/.

Page 17, line 264

Third, our study only included men aged ≥50 years and post-menopausal women; therefore, our results may not be applicable to younger men or premenopausal women.

2.The study adjusts for several confounders but may still be subjected to residual confounding. Is it possible to utilize advanced statistical methods such as propensity score matching or structural equation modeling to further control for potential confounding variables.

Reply: Thank you for your comment. In research, the choice of statistical methods depends on several factors, including the nature of the data and research questions.

1. Nature of the Data:

- Homogeneity: If the data lacks a clear distinction between treated and control groups, propensity score matching (PSM) cannot be applied because it requires such differentiation to match individuals based on propensity scores.

- Complexity: Structure equation modeling (SEM) might not be suitable if the focus is on adjusting for confounders rather than modeling complex relationships among latent and observed variables.

2. Research Objectives:

- Causal Inference vs. Adjustment: If the primary goal is to adjust for confounding variables to estimate the direct relationships between variables (e.g., between an independent variable and a dependent variable), multiple regression is straightforward and directly addresses this aim.

- Modeling Complex Relationships: SEM is more appropriate when the study aims to model and test complex theoretical frameworks involving latent variables, mediation, moderation, and causal pathways, which may not align with our current research goals.

Although PSM and SEM are powerful methods in certain contexts, they are not suitable for our study because our dataset lacks distinct treatment and control groups necessary for PSM. SEM is more appropriate when the study aims to model and test complex theoretical frameworks involving latent variables, mediation, moderation, and causal pathways, which may not align with our current research goals. Furthermore, our research goals primarily involve adjusting for confounders to estimate direct relationships between variables, aligning well with the capabilities of multiple regression. This method allows us to straightforwardly address our research questions while adhering to the assumptions and data structure of our study.

3.The study reports positive associations between fat indices and bone density, and negative associations with bone quality, but does not thoroughly explore these relationships. In discussion, please provide a more detailed analysis of the results, or discuss the biological mechanisms that might explain the differential effects of fat and lean indices on bone strength.

Reply: Thank you for your comment. The positive association between fat indices and bone density is attributed to the mechanical loading theory. However, accumulating evidence suggests a link between obesity and compromised bone quality. Various mechanisms are believed to explain how obesity negatively affects bone health, including alterations in bone-regulating hormones, inflammation, oxidative stress, and the endocannabinoid system, all of which impact bone cell metabolism. Additionally, both fat and lean mass contribute to bone density by increasing mechanical loading, and men naturally have a higher percentage of lean muscle mass than women. Therefore, lean mass contributes more to bone density in men than in women. In response to your insightful comments, we have further explored the association between fat indices and bone density and quality, and discussed the possible biological mechanisms. We have

---

## [Decision Letter · Decision Letter 1]

6 Sep 2024

PONE-D-24-20088R1Different contributions of fat and lean indices to bone strength by sexPLOS ONE

Dear Dr. Lin,

Thank you for submitting your manuscript to PLOS ONE. After careful consideration, we feel that it has merit but does not fully meet PLOS ONE’s publication criteria as it currently stands. Therefore, we invite you to submit a revised version of the manuscript that addresses the points raised during the review process.

We look forward to receiving your revised manuscript.

Kind regards,

Kiyoshi Sanada, PhD

Academic Editor

PLOS ONE

Reviewers' comments:

Reviewer's Responses to Questions

**Comments to the Author**

1. If the authors have adequately addressed your comments raised in a previous round of review and you feel that this manuscript is now acceptable for publication, you may indicate that here to bypass the “Comments to the Author” section, enter your conflict of interest statement in the “Confidential to Editor” section, and submit your "Accept" recommendation.

Reviewer #1: All comments have been addressed

Reviewer #2: (No Response)

2. Is the manuscript technically sound, and do the data support the conclusions?

Reviewer #1: Yes

Reviewer #2: Partly

3. Has the statistical analysis been performed appropriately and rigorously? 

Reviewer #1: Yes

Reviewer #2: Yes

4. Have the authors made all data underlying the findings in their manuscript fully available?

Reviewer #1: Yes

Reviewer #2: Yes

5. Is the manuscript presented in an intelligible fashion and written in standard English?

Reviewer #1: Yes

Reviewer #2: No

6. Review Comments to the Author

Reviewer #1: Dear Author

After moderate revision, I have identified two additional suggestions that could still benefit this article from further improvements.

1.Although the introduction has been improved, it could benefit from a clearer statement of the research gap and the novelty of the study. Explicitly highlight how this study differs from previous research and what new insights it offers.

2.Although the discussion section has been expanded, some results, such as the differences in how fat and lean indices contribute to bone strength in men and women, could be interpreted more thoroughly. Discuss the potential biological mechanisms in more detail and relate your findings more explicitly to the existing literatures.

Reviewer #2: Comments (Reviewer#2)

Thank you for the correction.

After the correction, the issue has been clarified. However, in order to improve the content, we would like to comment.

1, Minor Comment

The previous abstract was presented in the following sections: objectives (e.g. background), methods, results and conclusions, but no distinction is made in the revised version. Was this removed due to special instructions?

If not, it would be better to add the terms purpose (e.g. background), methods, results and conclusions.

2, Major Comment

Concerning the concluding part of the abstract, P2, L37-, and the concluding part of the text, P17, L269-.

The "wording and text in the conclusion of the abstract" does not adequately reflect the "conclusions of the main text". For example, in the conclusion of the main text, men show a relationship between HGS and BMD. However, the abstract conclusion describes a correlation between bone strength in men. The conclusion is the claim of this paper and is very important. The conclusion in the main text and the conclusion in the abstract do not have to be exactly the same sentence, but they should make the same claim.

7. PLOS authors have the option to publish the peer review history of their article (what does this mean?). If published, this will include your full peer review and any attached files.

Reviewer #1: No

Reviewer #2: No

---

## [Author Response · Author response to Decision Letter 1]

16 Sep 2024

1. If the authors have adequately addressed your comments raised in a previous round of review and you feel that this manuscript is now acceptable for publication, you may indicate that here to bypass the “Comments to the Author” section, enter your conflict of interest statement in the “Confidential to Editor” section, and submit your "Accept" recommendation.

Reviewer #1: All comments have been addressed

Reviewer #2: (No Response)

Reply: Thank you for your comment.

2. Is the manuscript technically sound, and do the data support the conclusions?

Reviewer #1: Yes

Reviewer #2: Partly

Reply: Thank you for your comment. We have revised the abstract to reflect the same claim found on page 2, line 37.

3. Has the statistical analysis been performed appropriately and rigorously?

Reviewer #1: Yes

Reviewer #2: Yes

Reply: Thank you for your comment.

4. Have the authors made all data underlying the findings in their manuscript fully available?

Reviewer #1: Yes

Reviewer #2: Yes

Reply: Thank you for your comment.

5. Is the manuscript presented in an intelligible fashion and written in standard English?

Reviewer #1: Yes

Reviewer #2: No

Reply: Thank you for your comment. Our manuscript has undergone an additional round of English editing to enhance the language and improve clarity and understanding.

6. Review Comments to the Author

Reviewer #1: Dear Author

After moderate revision, I have identified two additional suggestions that could still benefit this article from further improvements.

1.Although the introduction has been improved, it could benefit from a clearer statement of the research gap and the novelty of the study. Explicitly highlight how this study differs from previous research and what new insights it offers.

Reply: Thank you for your comment. We have included a statement of the research gap on page 5, line 92, to emphasize how our study differs from previous research.

Page 5, line 92.

Therefore, body composition indices varied according to sex, and muscle strength was a key measure of lean indices. Bone strength was defined as the integration of bone density and quality. To the best of our knowledge, no previous study has comprehensively investigated the association between different fat and lean indices with bone strength in men and women. Therefore, we aimed to elucidate the effects of the lean and fat indices on bone strength in men and women.

2.Although the discussion section has been expanded, some results, such as the differences in how fat and lean indices contribute to bone strength in men and women, could be interpreted more thoroughly. Discuss the potential biological mechanisms in more detail and relate your findings more explicitly to the existing literatures.

Reply: Thank you for your comment. We have discussed the potential biological mechanisms in greater detail on page 17, line 260.

Page 17, line 260.

The different contributions of fat and muscle to bone health in men and women are primarily driven by biological mechanisms related to hormonal regulation, mechanical loading, and the distinct ways in which fat and muscle interact with bone tissue. The muscle includes mechanostat function, in which muscles apply stress to bones through contractions during movement, triggering bone remodeling. This mechanical loading is detected by osteocytes, which then signal new bone formation (osteogenesis) to reinforce the skeletal structure [25, 26]. Men typically have greater muscle mass, leading to greater mechanical loading on bones. This results in a stronger stimulus for bone formation and an increased bone mineral density. In contrast, women tend to have a lower muscle mass, which results in less mechanical loading and a weaker stimulus for bone remodeling. This can contribute to relatively lower bone mineral density, particularly after menopause, when muscle mass may further decline. Second, muscles release signaling molecules called myokines that influence bone metabolism. Higher muscle mass in men leads to greater secretion of myokines such as insulin-like growth factor-1, which promotes both muscle growth and bone formation [28, 29]. Although women also benefit from myokine release, their lower muscle mass results in a reduced contribution of this pathway to bone-building processes. The role of fat includes mechanical loading from body weight. Like muscle mass, increased fat mass exerts mechanical loading on bones, stimulating bone formation [11]. Although this effect is less direct and efficient than muscle-induced mechanical loading, body weight from fat still contributes to bone health. Men generally have less body fat than women, leading to less mechanical loading from fat. In contrast, higher body fat in women contributes more significantly to mechanical loading and may positively influence bone mineral density. However, excessive fat can have negative effects, as it may cause systemic inflammation [27]. Second, increased fat-derived hormones (adipokines) such as leptin produced by increased adipose tissue seem to have a protective effect on bone tissue [30]. Leptin plays a less significant role in maintaining bone mineral density in men because of their lower fat mass, whereas it plays a more substantial role in women because of their higher fat mass [31, 32]. The complex interplay between muscle, fat, and hormones affects bone health differently in men and women, with hormonal differences being central to modulating the effect of fat and muscle on bone health.

Reviewer #2: Comments (Reviewer#2)

Thank you for the correction.

After the correction, the issue has been clarified. However, in order to improve the content, we would like to comment.

1, Minor Comment

The previous abstract was presented in the following sections: objectives (e.g. background), methods, results and conclusions, but no distinction is made in the revised version. Was this removed due to special instructions?

If not, it would be better to add the terms purpose (e.g. background), methods, results and conclusions.

Reply: Thank you for your comment. The editors requested that our manuscript comply with PLOS ONE's style guidelines, which require an unstructured abstract.

The PLOS ONE style templates can be found at 

2, Major Comment

Concerning the concluding part of the abstract, P2, L37-, and the concluding part of the text, P17, L269-.

The "wording and text in the conclusion of the abstract" does not adequately reflect the "conclusions of the main text". For example, in the conclusion of the main text, men show a relationship between HGS and BMD. However, the abstract conclusion describes a correlation between bone strength in men. The conclusion is the claim of this paper and is very important. The conclusion in the main text and the conclusion in the abstract do not have to be exactly the same sentence, but they should make the same claim.

Reply: Thank you for your comment. We have revised the abstract to reflect the same claim found on page 2, line 37.

Page 2, line 37.

Fat indices contributed more to bone mineral density in women, whereas lean indices were more positively associated with bone strength in men. Considering different fat indices, appendicular lean mass was more strongly associated with bone mineral density in women, whereas hand grip strength contributed more to bone mineral density in men. In other words, maintaining muscle mass and strength is equally important for bone health.

7. PLOS authors have the option to publish the peer review history of their article (what does this mean?). If published, this will include your full peer review and any attached files.

Do you want your identity to be public for this peer review? For information about this choice, including consent withdrawal, please see our Privacy Policy.

Reviewer #1: No

Reviewer #2: No

Reply: Thank you for your comment.

---

## [Decision Letter · Decision Letter 2]

27 Sep 2024

PONE-D-24-20088R2Different contributions of fat and lean indices to bone strength by sexPLOS ONE

Dear Dr. Lin,

Thank you for submitting your manuscript to PLOS ONE. After careful consideration, we feel that it has merit but does not fully meet PLOS ONE’s publication criteria as it currently stands. Therefore, we invite you to submit a revised version of the manuscript that addresses the points raised during the review process.

We look forward to receiving your revised manuscript.

Kind regards,

Kiyoshi Sanada, PhD

Academic Editor

PLOS ONE

Journal Requirements:

Reviewers' comments:

Reviewer's Responses to Questions

**Comments to the Author**

1. If the authors have adequately addressed your comments raised in a previous round of review and you feel that this manuscript is now acceptable for publication, you may indicate that here to bypass the “Comments to the Author” section, enter your conflict of interest statement in the “Confidential to Editor” section, and submit your "Accept" recommendation.

Reviewer #1: All comments have been addressed

Reviewer #2: All comments have been addressed

2. Is the manuscript technically sound, and do the data support the conclusions?

Reviewer #1: Yes

Reviewer #2: Yes

3. Has the statistical analysis been performed appropriately and rigorously? 

Reviewer #1: Yes

Reviewer #2: Yes

4. Have the authors made all data underlying the findings in their manuscript fully available?

Reviewer #1: Yes

Reviewer #2: Yes

5. Is the manuscript presented in an intelligible fashion and written in standard English?

Reviewer #1: Yes

Reviewer #2: Yes

6. Review Comments to the Author

Reviewer #1: Dear Author

The author revised the abstract to align with the main text, ensuring consistency in the conclusions drawn about handgrip strength (HGS) and bone mineral density (BMD) between men and women. The inclusion of a more detailed explanation about the different contributions of fat and lean indices also provides greater clarity to the biological mechanisms behind the sex differences in bone strength

After 2nd revision, there are still minor points worth noting:

1.While the revised abstract now mirrors the conclusions in the text, it could still benefit from being more concise. The distinction between fat indices and lean indices, as well as their contributions to bone health, is now clearer, but simplifying the language in the abstract would improve readability.

2.The expanded discussion about potential biological mechanisms is comprehensive, but a more focused discussion on the most relevant mechanisms could streamline this section.

Reviewer #2: (No Response)

7. PLOS authors have the option to publish the peer review history of their article (what does this mean?). If published, this will include your full peer review and any attached files.

Reviewer #1: **Yes: **Robert Wen-Wei Hsu

Reviewer #2: No

---

## [Author Response · Author response to Decision Letter 2]

2 Oct 2024

1. If the authors have adequately addressed your comments raised in a previous round of review and you feel that this manuscript is now acceptable for publication, you may indicate that here to bypass the “Comments to the Author” section, enter your conflict of interest statement in the “Confidential to Editor” section, and submit your "Accept" recommendation.

Reviewer #1: All comments have been addressed

Reviewer #2: All comments have been addressed

Reply: Thank you for your comment.

2. Is the manuscript technically sound, and do the data support the conclusions?

Reviewer #1: Yes

Reviewer #2: Yes

Reply: Thank you for your comment.

3. Has the statistical analysis been performed appropriately and rigorously?

Reviewer #1: Yes

Reviewer #2: Yes

Reply: Thank you for your comment.

4. Have the authors made all data underlying the findings in their manuscript fully available?

Reviewer #1: Yes

Reviewer #2: Yes

Reply: Thank you for your comment.

5. Is the manuscript presented in an intelligible fashion and written in standard English?

Reviewer #1: Yes

Reviewer #2: Yes

Reply: Thank you for your comment.

6. Review Comments to the Author

Reviewer #1: Dear Author

The author revised the abstract to align with the main text, ensuring consistency in the conclusions drawn about handgrip strength (HGS) and bone mineral density (BMD) between men and women. The inclusion of a more detailed explanation about the different contributions of fat and lean indices also provides greater clarity to the biological mechanisms behind the sex differences in bone strength

After 2nd revision, there are still minor points worth noting:

1.While the revised abstract now mirrors the conclusions in the text, it could still benefit from being more concise. The distinction between fat indices and lean indices, as well as their contributions to bone health, is now clearer, but simplifying the language in the abstract would improve readability.

Reply: Thank you for your comment. The abstract has undergone an additional round of English editing to enhance the language and improve clarity and understanding.

Page 2, line 20.

ABSTRACT

Bone strength depends on both bone density and quality. However, the impact of body composition indices on bone strength in men and women remains unclear. This study aimed to investigate the association of different fat and lean indices with bone strength according to sex. In this cross-sectional study involving 1,419 participants, bone mineral density (BMD) and body composition were measured using dual-energy X-ray absorptiometry. Bone quality was assessed using the trabecular bone score (TBS). Fat indices included total fat mass, body fat percentage, and waist circumference, while lean indices included appendicular lean mass (ALM) and hand grip strength. All fat indices demonstrated a positive association with BMD and a negative association with TBS in both men and women. Fat indices were more closely associated with BMD in women than in men. Furthermore, lean indices contributed more to BMD in men than in women. In women, ALM contributed more to BMD than hand grip strength, whereas in men, hand grip strength had a greater impact on BMD than ALM. Hand grip strength was also positively associated with TBS in men. Overall, fat indices had a greater influence on BMD in women, while lean indices were more positively associated with bone strength in men. Considering different fat indices, ALM was more closely associated with BMD in women, whereas hand grip strength played a greater role in men. Thus, maintaining both muscle mass and strength is crucial for preserving bone mass.

2.The expanded discussion about potential biological mechanisms is comprehensive, but a more focused discussion on the most relevant mechanisms could streamline this section.

Reply: Thank you for your comment. We have focused on the most relevant mechanisms in the discussion section on page 17, line 260. 

Page 17, line 260.

The different contributions of fat and muscle to bone health in men and women are primarily driven by biological mechanisms related to mechanical loading and the distinct ways in which fat and muscle interact with bone tissue. The muscle includes mechanostat function, in which muscles apply stress to bones through contractions during movement, triggering bone remodeling. This mechanical loading is detected by osteocytes, which then signal new bone formation (osteogenesis) to reinforce the skeletal structure [25, 26]. Men typically have greater muscle mass, leading to greater mechanical loading on bones. This results in a stronger stimulus for bone formation and an increased BMD. In contrast, women tend to have a lower muscle mass, which results in less mechanical loading and a weaker stimulus for bone remodeling. This can contribute to relatively lower BMD, particularly after menopause, when muscle mass may further decline. The role of fat includes mechanical loading from body weight. Like muscle mass, increased fat mass exerts mechanical loading on bones, stimulating bone formation [11]. Although this effect is less direct and efficient than muscle-induced mechanical loading, body weight from fat still contributes to bone health. Men generally have less body fat than women, leading to less mechanical loading from fat. In contrast, higher body fat in women contributes more significantly to mechanical loading and may positively influence BMD. However, excessive fat can have negative effects, as it may cause systemic inflammation [27]. The complex interplay between muscle and fat affects bone health differently in men and women, with hormonal differences modulating the effect of fat and muscle on bone health [25, 26].

Reviewer #2: (No Response)

Reply: Thank you for your comment.

7. PLOS authors have the option to publish the peer review history of their article (what does this mean?). If published, this will include your full peer review and any attached files.

Do you want your identity to be public for this peer review? For information about this choice, including consent withdrawal, please see our Privacy Policy.

Reviewer #1: Yes: Robert Wen-Wei Hsu

Reviewer #2: No

Reply: Thank you for your comment.

---

## [Decision Letter · Decision Letter 3]

23 Oct 2024

PONE-D-24-20088R3Different contributions of fat and lean indices to bone strength by sexPLOS ONE

Dear Dr. Lin,

Thank you for submitting your manuscript to PLOS ONE. After careful consideration, we feel that it has merit but does not fully meet PLOS ONE’s publication criteria as it currently stands. Therefore, we invite you to submit a revised version of the manuscript that addresses the points raised during the review process.

We look forward to receiving your revised manuscript.

Kind regards,

Kiyoshi Sanada, PhD

Academic Editor

PLOS ONE

Journal Requirements:

Reviewers' comments:

Reviewer's Responses to Questions

**Comments to the Author**

1. If the authors have adequately addressed your comments raised in a previous round of review and you feel that this manuscript is now acceptable for publication, you may indicate that here to bypass the “Comments to the Author” section, enter your conflict of interest statement in the “Confidential to Editor” section, and submit your "Accept" recommendation.

Reviewer #1: All comments have been addressed

Reviewer #3: All comments have been addressed

2. Is the manuscript technically sound, and do the data support the conclusions?

Reviewer #1: Yes

Reviewer #3: Yes

3. Has the statistical analysis been performed appropriately and rigorously? 

Reviewer #1: Yes

Reviewer #3: Yes

4. Have the authors made all data underlying the findings in their manuscript fully available?

Reviewer #1: Yes

Reviewer #3: Yes

5. Is the manuscript presented in an intelligible fashion and written in standard English?

Reviewer #1: Yes

Reviewer #3: Yes

6. Review Comments to the Author

Reviewer #1: Dear Author:

The authors addressed key concerns by aligning the abstract with the findings, refining biological mechanism discussions, and checking for multicollinearity. However, further streamlining of the abstract and organization of the discussion would enhance clarity. A post-hoc power analysis and exploration of interaction effects are still needed to strengthen statistical rigor.

The authors appropriately acknowledge limited generalizability and suggest future studies. With minor revisions—focusing on clarity, statistical robustness, and data availability—the manuscript will be ready for publication and provide valuable insights into bone health differences across sexes.

Reviewer #3: Osteoporosis is recognized as a serious problem that is associated with the need for nursing care and threatens healthy life expectancy. This study can be applied to simple prediction of regional BMD related to fractures in adults in the Asian region. It is expected that this method will be used in primary care in the future. The review comments to date have been fully addressed.

7. PLOS authors have the option to publish the peer review history of their article (what does this mean?). If published, this will include your full peer review and any attached files.

Reviewer #1: **Yes: **Robert Wen-Wei Hsu

Reviewer #3: No

---

## [Author Response · Author response to Decision Letter 3]

29 Oct 2024

Comments to the Author

1. If the authors have adequately addressed your comments raised in a previous round of review and you feel that this manuscript is now acceptable for publication, you may indicate that here to bypass the “Comments to the Author” section, enter your conflict of interest statement in the “Confidential to Editor” section, and submit your "Accept" recommendation.

Reviewer #1: All comments have been addressed

Reviewer #3: All comments have been addressed

Reply: Thank you for your comment.

2. Is the manuscript technically sound, and do the data support the conclusions?

Reviewer #1: Yes

Reviewer #3: Yes

Reply: Thank you for your comment.

3. Has the statistical analysis been performed appropriately and rigorously?

Reviewer #1: Yes

Reviewer #3: Yes

Reply: Thank you for your comment.

4. Have the authors made all data underlying the findings in their manuscript fully available?

Reviewer #1: Yes

Reviewer #3: Yes

Reply: Thank you for your comment.

5. Is the manuscript presented in an intelligible fashion and written in standard English?

Reviewer #1: Yes

Reviewer #3: Yes

Reply: Thank you for your comment.

6. Review Comments to the Author

Reviewer #1: Dear Author:

The authors addressed key concerns by aligning the abstract with the findings, refining biological mechanism discussions, and checking for multicollinearity. However, further streamlining of the abstract and organization of the discussion would enhance clarity. A post-hoc power analysis and exploration of interaction effects are still needed to strengthen statistical rigor.

The authors appropriately acknowledge limited generalizability and suggest future studies. With minor revisions—focusing on clarity, statistical robustness, and data availability—the manuscript will be ready for publication and provide valuable insights into bone health differences across sexes.

Reply: We appreciate your valuable comments to help improve the quality of our manuscript. In response, we have not only conducted an additional round of English editing to enhance clarity and comprehension, but also strengthened the statistical robustness of our analysis on page 8, line 151 and page 12, line 193. Our responses to the reviewers' comments are provided below, and the revised manuscript has been updated accordingly. Additionally, the dataset is owned by the IRB of Cheng Hsin General Hospital. The IRB approved the data analysis for our study but did not approve data sharing. Therefore, we do not have permission to share the dataset. Interested researchers can request access to the data by contacting the IRB of Cheng Hsin General Hospital at chghirb@chgh.org.tw. Access to the data will be granted in the same manner as it was for the authors, who do not have any special access privileges. We have included the data availability statement on page 20, line 317. We hope that the revised manuscript will have an enhanced scientific impact.

Page 8, line 151

An interaction model was built to investigate the statistical interaction between body composition indices and sex. A post-hoc power analysis based on an alpha level (0.05) was performed to strengthen the statistical robustness. 

Page 12, line 187

HGS was more positively associated with BMDs and the TBS in men than in women. In contrast, HGS showed minimal association with total hip BMD in women (Table 3). Additionally, there was significant interaction between HGS and sex. The VIFs in all models were less than 10, indicating a low degree of multicollinearity. As with women, the R2 values in men for spine BMD and the TBS were low, but they were high for both femoral neck and total hip BMDs. In contrast, when models were adjusted for HGS, the R2 values were higher in men than in women. A post-hoc power analysis showed that the power exceeded 80%.

Page 18, line 290

Nevertheless, our study had some limitations that must be considered when interpreting the results. First, this was a cross-sectional study; therefore, the causal inferences could not be made. Second, the participants belonged exclusively to the Asian population, which may have affected the generalizability of our results. Third, our study only included men aged ≥50 years and post-menopausal women; therefore, our results may not be applicable to younger men or premenopausal women. Further studies across various populations were needed to confirm our results.

Page 20, line 317

Data Availability statement: 

The data set is owned by the institutional review board of Cheng Hsin General Hospital. The institutional review board of Cheng Hsin General Hospital only approved the data analysis in our study and did not approve data sharing. Therefore, we do not have permission to share the data set. Interested researchers can submit data access requests to the institutional review board of Cheng Hsin General Hospital through the following email address: chghirb@chgh.org.tw Others would be able to access the data in the same manner as the authors. 

Reviewer #3: Osteoporosis is recognized as a serious problem that is associated with the need for nursing care and threatens healthy life expectancy. This study can be applied to simple prediction of regional BMD related to fractures in adults in the Asian region. It is expected that this method will be used in primary care in the future. The review comments to date have been fully addressed.

Reply: Thank you for your comment.

7. PLOS authors have the option to publish the peer review history of their article (what does this mean?). If published, this will include your full peer review and any attached files.

Do you want your identity to be public for this peer review? For information about this choice, including consent withdrawal, please see our Privacy Policy.

Reviewer #1: Yes: Robert Wen-Wei Hsu

Reviewer #3: No

Reply: Thank you for your comment.

---

## [Editor Report · Decision Letter 4]

31 Oct 2024

Different contributions of fat and lean indices to bone strength by sex

PONE-D-24-20088R4

Dear Dr. Lin,

We’re pleased to inform you that your manuscript has been judged scientifically suitable for publication and will be formally accepted for publication once it meets all outstanding technical requirements.

Kind regards,

Kiyoshi Sanada, PhD

Academic Editor

PLOS ONE
---

## [Editor Report · Acceptance letter]

5 Nov 2024

PONE-D-24-20088R4 

PLOS ONE

Dear Dr. Lin, 

I'm pleased to inform you that your manuscript has been deemed suitable for publication in PLOS ONE. Congratulations! Your manuscript is now being handed over to our production team.

Kind regards, 

on behalf of

Dr. Kiyoshi Sanada 

Academic Editor

PLOS ONE